# Enhanced Lithium Storage Performance of α-MoO_3_/CNTs Composite Cathode

**DOI:** 10.3390/nano13152272

**Published:** 2023-08-07

**Authors:** Dawei Sheng, Ang Gao, Xiaoxu Liu, Qiang Zhang

**Affiliations:** 1Key Laboratory for Photonic and Electronic Bandgap Materials, Ministry of Education, School of Physics and Electronic Engineering, Harbin Normal University, Harbin 150025, China; tsongyi@163.com (D.S.); ga19990528@163.com (A.G.); 2Shaanxi Key Laboratory of Green Preparation and Functionalization for Inorganic Materials, School of Materials Science and Engineering, Shaanxi University of Science and Technology, Xi’an 710021, China

**Keywords:** cathode, α-MoO_3_, carbon nanotubes, lithium-ion batteries

## Abstract

Orthorhombic molybdenum oxide (α-MoO_3_), as a one-layered pseudocapacitive material, has attracted widespread attention due to its high theoretical lithium storage specific capacity (279 mAh/g) for lithium-ion batteries’ cathode. Nevertheless, low conductivity, slack reaction kinetics, and large volume change during Li^+^ ions intercalation and deintercalation seriously limit the practical application of α-MoO_3_. Herein, we added a small number of CNTs (1.76%) to solve these problems in a one-step hydrothermal process for preparing the α-MoO_3_/CNTs composite. Because of the influence of CNTs, the α-MoO_3_ nanobelt in the α-MoO_3_/CNTs composite had a larger interlayer spacing, which provided more active sites and faster reaction kinetics for lithium storage. In addition, CNTs formed a three-dimensional conductive network between α-MoO_3_ nanobelts, enhanced the electrical conductivity of the composite, accelerated the electron conduction, shortened the ion transport path, and alleviated the structural fragmentation caused by the volume expansion during the α-MoO_3_ intercalation and deintercalation of Li^+^ ions. Therefore, the α-MoO_3_/CNTs composite cathode had a significantly higher rate performance and cycle life. After 150 cycles, the pure α-MoO_3_ cathode had almost no energy storage, but α-MoO_3_/CNTs composite cathode still retained 93 mAh/g specific capacity.

## 1. Introduction

Nowadays, to cope with the ever-increasing energy demands, it is imperative to develop energy storage technologies with low cost, high capacity, and long cycle life [1]. In typical energy storage technologies, rechargeable lithium-ion batteries (LIBs) stand out for their environmental friendliness, high energy density, high operating voltage, and no memory effect. However, with the rapid development of portable electronic devices, electric vehicles, grid applications, etc., researchers should continue to dive into the exploration of higher-capacity electrode materials to meet the increasing needs for higher energy-density storage, although LiCoO_2_, LiMn_2_O_4_, LiFePO_4_, and LiNi_0.33_Co_0.33_Mn_0.33_O_2_ cathode materials have been commercialized [2]. Transition metal oxides with high reversible capacity and excellent electrochemical performance based on pseudocapacitive energy storage mechanisms are good candidates for cathode materials [3]. As a cathode material of LIBs, MoO_3_ has piqued broad research interests due to its abundant resources, stable chemical properties, and high theoretical specific capacity [4,5,6,7]. Taking the widely studied thermodynamically stable orthorhombic stable a-phase MoO_3_ (α-MoO_3_) with anisotropic [5] as an example, its unique layered structure consists of two edge-sharing layers of [MoO_6_] octahedra sharing corners along the crystal orientation [001] and [100], and a two-dimensional structure stacked by van der Waals forces along [010]. This two-dimensional structure of α-MoO_3_ allows ions (such as Li^+^) to intercalate into the interlayer, leading to highly redox active Mo^6+^/Mo^5+^ and Mo^5+^/Mo^4+^ double electron reactions, with a high specific capacity of nearly 300 mAh/g [8]. In our previous work on α-MoO_3_ as an anode for Li^+^ storage based on conversion reactions, the intercalation reaction of α-MoO_3_ as a cathode had better structural stability and a faster kinetic reaction [9]. In recent studies, xia et al. monitored the structural changes of MoO_3_ during lithium storage by in situ TEM, providing intuitive evidence for its lithiation and delithiation behavior [10]. The MoO_3_ cathode modified by ammoniation by Wang et al. showed good stability in the voltage of 1–3.5 V [8]. The electrodeposition of MoO_3_ nanoparticles on conductive materials by Zhang et al. showed excellent kinetics [11]. However, the inherent low conductivity and poor rate performance of α-MoO_3_ as a transition metal oxide still limit its application [12]. In addition, during the initial cycle, an irreversible phase transition occurs near the discharge platform at 2.8 V, which causes α-MoO_3_ layer deformation and the capacity to rapidly decay in subsequent cycles. To solve these problems, researchers have enhanced pseudocapacitance storage by constructing nanostructures and manufacturing oxygen vacancies, electrical conductivity by pre-intercalation and compounding with carbon materials, the electrochemical performance of α-MoO_3_, etc. [13,14,15,16]. Although the lithium storage performance of α-MoO_3_ has been improved, the preparation methods are often complicated, and the improvement of lithium storage performance is still insufficient. For example, the fabrication of oxygen defects requires strict control of the etching time. The pre-intercalation is necessary for controlling the amount of ion insertion and prevention of detachment during subsequent cycles. Several studies have indicated that ensuring the uniformity of composite materials when compounding with carbon materials is difficult [11,13,17,18,19,20,21,22,23]. Therefore, it is still challenging to effectively improve the electrochemical performance of α-MoO_3_ of lithium storage to meet future large-scale production needs.

Herein, unique α-MoO_3_/carbon nanotubes (CNTs) composite cathode was synthesized by adding a small amount of functionalized CNTs (1.76%) in a one-step hydrothermal process, and we used hydrophobic multiwall carbon nanotubes (CNTs). To facilitate the synthesis of α-MoO_3_/CNTs composite with an internal embedded structure, we treated the CNTs to make them rich in functional groups. Specifically, we acidified the multiwall CNTs to make them hydrophilic. Given the influence of CNTs, the α-MoO_3_ nanobelt in the α-MoO_3_/CNTs composite introduced oxygen vacancies and had a larger interlayer spacing, providing more active sites and faster kinetic reaction for lithium storage. In addition, the CNTs formed a three-dimensional conductive network between α-MoO_3_ nanobelts, enhancing the composite’s electrical conductivity, accelerating the electron conduction, shortening the ion transport path, and alleviating the structural fragmentation caused by the volume expansion in the Li^+^ storage process. Thus, the rate performance and the cycle life of α-MoO_3_/CNTs composite were significantly improved compared to pure α-MoO_3_.

## 2. Results and Discussion

### 2.1. Structure and Morphology Analysis

The α-MoO_3_/CNTs composite was prepared by a simple one-step hydrothermal method. Figure 1a shows the effect of CNTs on α-MoO_3_ in the α-MoO_3_/CNTs composites, including the expanded interlayer distance α-MoO_3_ structure and CNTs’ three-dimensional conductive network in the α-MoO_3_/CNTs composite. This point is confirmed by the X-ray diffraction (XRD) results of α-MoO_3_/CNTs composite and pure α-MoO_3_. As shown in Figure 1b, the X-ray excitation source was monochrome Al Ka (hv = 1486.6 eV), power 150 W, X-ray beam spot 500 μm, and an energy analyzer through an energy of 30 eV. Due to the low content of CNTs, the α-MoO_3_/CNTs composite corresponds well to orthorhombic α-MoO_3_ (JCPDS No. 05-0508, space group: Pbnm (62)) [24,25,26]. The α-MoO_3_/CNTs composite had better crystallinity without other impurity phases, and the (0k0) peak was very sharp, which further illustrates that the layered structure of α-MoO_3_ held together along the b-axis and preferred orientation growth along the [001] direction. This is consistent with the long nanobelt shape in the SEM image. The SEM image (Figure 1c) shows a thin and uniform nanobelt structure of the α-MoO_3_/CNTs composite with a length of about 1–6 μm [27,28,29,30]. Compared with pure α-MoO_3_, the composite nanobelts were shorter, and the TEM image in Figure 1d indicates the presence of bent CNTs between the α-MoO_3_ nanobelts. The CNTs were uniformly distributed between the α-MoO_3_ nanobelts to form a serviceable conductive network, which greatly increased the conductivity during the energy storage process. It is worth noting that the XRD patterns were partially enlarged, as shown in Figure 1e. Compared with the (0k0) peak position of pure α-MoO_3_, the peak position of the CNTs added α-MoO_3_/CNTs composite was significantly shifted to a smaller angle. This verifies that the CNTs embedded α-MoO_3_ in the α-MoO_3_/CNTs composite had a larger interlayer spacing than the pure α-MoO_3_.

Figure 2a is the Raman spectra comparison of the α-MoO_3_/CNTs composite and pure α-MoO_3_. Both the D-peak and G-peak are the Raman characteristic peaks of C-atom crystals, which were observed around 1300 cm^−1^ and 1600 cm^−1^, respectively. The D and G bands in the Raman spectra can verify the presence of a small number of CNTs in the α-MoO_3_/CNTs composite. The characteristic bands of α-MoO_3_ in the composite were observed at 1006 cm^−1^ (A_g_, ν_as_ Mo=O stretch) and 828 cm^−1^ (A_g_, ν_as_ Mo=O stretch), which corresponds to the axially symmetric stretching vibration of the terminal Mo=O along the a-axis and b-axis, and the characteristic bands of α-MoO_3_ were observed at 676 cm^−1^ (B_2g_, B_3g_, ν_as_ Mo-O-Mo stretch) and 482 cm^−1^ (A_g_, ν_as_ Mo-O-Mo stretch and bend), which corresponded to the bridge oxygen bond with weakly bound oxygen along the c-axis. The peak on the right side of Figure 2a is the D-band and G-band of the CNTs in the α-MoO_3_/CNTs composite, and the D-band and G-band frequencies were 1373 and 1620 cm^−1^, respectively. The *I_D_*/*I_G_* = 0.876, indicating that the disorder of CNTs was high, which is consistent with the TEM image of Figure 1d. To determine the content of CNTs in the α-MoO_3_/CNTs composite, an thermogravimetric analysis of the composites was carried out. Figure 1b shows the TG and DTG curves of the α-MoO_3_/CNTs composite at a 10 °C/min heating rate in an air atmosphere. The apparent weight loss from 150 °C to 450 °C was the reaction of bound water in the composites, and the apparent weight loss from 450 °C to 550 °C could be ascribed to the decomposition of CNTs. Therefore, the content of CNTs was 1.76%, and a small amount of CNTs greatly influenced the structure of α-MoO_3_ in the α-MoO_3_/CNTs composite. The XPS spectra of the α-MoO_3_/CNTs composite and pure α-MoO_3_ are shown in Figure 2c,d. The XPS survey spectrum signifies the coexistence of Mo, O, and C elements in the α-MoO_3_/CNTs composite. The high-resolution spectrum of Mo 3d in the composite (Appendix A) shows a pair of Mo^6+^ peaks and a pair of weaker Mo^5+^ peaks. The 232.8 and 235.95 eV peaks belonged to Mo 3d_5/2_ and Mo 3d_3/2_ peaks of Mo^6+^. The 231.7 and 234.85 eV peaks corresponded to Mo 3d_5/2_ and Mo 3d_3/2_ peaks of Mo^5+^. Because of the presence of Mo^5+^, the α-MoO_3_ component in the α-MoO_3_/CNTs composite contains oxygen defected. Compared to pure α-MoO_3_, the peak position was almost not shifted, but the Mo^5+^/Mo^6+^ of α-MoO_3_/CNTs composite was more extensive because CNTs made the synthesized α-MoO_3_ contain more defects. The value of Mo^5+^ and Mo^6+^ was 0.072, so the calculation of α-MoO_3−x_ with defects was x = 0.03 in the composite. The strong peak at 530.6 eV in the O 1s high-resolution spectrum of Appendix A corresponded to the Mo-O bond in α-MoO_3_, while 531 eV corresponded to the C=O bond. The content of the C=O bond in the composite was lower due to the presence of CNTs. The C 1s high-resolution spectra Figure 2d of α α-MoO_3_/CNTs composite can be fitted into three peaks of 284.6, 285.1, and 288.75 eV, corresponding to C=C, C-C, and C=O bonds, respectively. The high content of the C-C bond was from the CNTs.

Due to the addition of CNTs in the hydrothermal reaction, some oxygen defects were introduced during the synthesis of α-MoO_3_. The α-MoO_3_ had a larger interlayer spacing but still maintained the nanobelt structure in the α-MoO_3_/CNTs composite. If the content of CNTs is increased, the [MoO_6_] octahedra structure will be more distorted and the nanobelts will be broken during the synthesis process. The α-MoO_3_/CNTs composite synthesized by adding a small number of CNTs in a one-step hydrothermal process is quite different from the microstructure of pure α-MoO_3_ and will have a greater effective effect on the energy storage process.

### 2.2. Electrochemical Analysis

To further study the effect of CNTs on α-MoO_3_ as a cathode material for lithium-ion batteries, the first three galvanostatic charge/discharge measurements of the α-MoO_3_/CNTs composite cathode and the pure α-MoO_3_ cathode were performed at a current density of 50 mA/g between the potential of 1 and 3.25 V. The results were contrasted with cases from the other work, as shown in Table 1. From the galvanostatic charge/discharge curves of the α-MoO_3_/CNTs composite cathode in Figure 3a, it can be seen that there was a discharge voltage plateau at about 2.75 V in the first cycle. The plateaus corresponded to Li^+^ intercalated in α-MoO_3_ to form a Li_x_MoO_3_ solid solution, and these discharge plateaus disappeared in the subsequent cycles. The Li^+^ were reversibly intercalated and deintercalated in the α-MoO_3_/CNTs composite cathode, corresponding to the discharge voltage plateau at 2.25 V and the charging voltage plateau at 2.60 V, respectively. The initial discharge-specific capacity was 296 mAh/g, and the initial coulombic efficiency (ICE) reached 85%. The energy storage was primarily carried out by inserting and extracting Li^+^ by α-MoO_3_ in a composite cathode. In addition, the capacity of the pure α-MoO_3_ cathode without CNTs constantly decayed from Figure 3c, indicating that the α-MoO_3_/CNTs composite had better electrochemical performance. At the same current density, the first discharge-specific capacity of the pure α-MoO_3_ cathode was 268 mAh/g, but the first-coulomb efficiency was only 68%, which is probably the reason that the structure of pure α-MoO_3_ was irreversibly destroyed during the intercalation of Li^+^ [11,31]. In contrast, the structural failure of the α-MoO_3_/CNTs composite was caused by the volume expansion of α-MoO_3_ during Li^+^ storage [20]. Because of the embedding of CNTs, the α-MoO_3_ nanobelt in the α-MoO_3_/CNTs composite had a larger interlayer spacing, which provided more active sites of Li^+^ intercalation and deintercalation. Comparing the irreversible Li^+^ storage part of the α-MoO_3_/CNTs composite cathode with that of pure α-MoO_3_ by the galvanostatic charge/discharge curves, the lithium storage process of α-MoO_3_ by the Li^+^ intercalation and deintercalation of the [MoO_6_] octahedral inter-layers and intra-layers positions, as well as the irreversible Li^+^ storage, can be attributed to the Li^+^ intercalated in the α-MoO_3_ intra-layer positions, which could not be completely reversibly extracted [13,24]. The α-MoO_3_/CNTs composite cathode corresponded to only irreversible 0.23 Li^+^/Mo, while pure α-MoO_3_ corresponded to irreversible 0.45 Li^+^/Mo. The pure α-MoO_3_ corresponded to more irreversible Li^+^ ion intercalation, which further verifies the improvement of CNTs on the reversible Li^+^ ions storage part of α-MoO_3_. Comparing the cyclic voltammetry curves of the first three cycles at a scan rate of 0.2 mV/s of the α-MoO_3_/CNTs composite cathode with the pure α-MoO_3_ cathode, Figure 3b shows that the reduction peaks of the α-MoO_3_/CNTs composite cathode appeared at 2.7 and 2.2 V. However, the reduction peak at 2.7 V disappeared in the subsequent cycle, which was caused by the irreversible intercalation of Li^+^ into α-MoO_3_ in α-MoO_3_/CNTs composite cathode and the generation of the interface phase between the composite cathode and the electrolyte. The results correspond to the galvanostatic charge-discharge curve. As Figure 3d shows, the reduction peaks of the pure α-MoO_3_ cathode appeared at 2.6, 2.25, and 1.95 V, respectively, and then the peak position shifted significantly in the subsequent cycle, and the reduction peak appeared only at 2.1 V. These may have been due to the serious collapse of the layered structure of pure α-MoO_3_ cathode after the first Li^+^ insertion, and the reduction and instability of Li^+^ intercalation active sites between and within layers of pure α-MoO_3_ [32]. Compared with pure α-MoO_3_, the gap between the oxidation peak and the reduction peak was smaller, and the peak current was even larger in the CNTs added α-MoO_3_/CNTs composite cathode, indicating that the polarization phenomenon was smaller and the kinetic reaction was faster, respectively. 

To explore the Li^+^ storage mechanism of the CNTs added to the α-MoO_3_/CNTs composite cathode, we characterized the morphology and elemental chemical state of the Li^+^ ion-intercalated α-MoO_3_/CNTs composite cathode after the first discharge. Figure 4a shows the TEM image of α-MoO_3_/CNTs first discharge to 1 V at a current density of 100 mA/g. It can be seen that the CNTs were uniformly distributed near the α-MoO_3_ nanobelts. The intercalation of Li^+^ during discharge increases the width of the α-MoO_3_ nanobelts of composite, according to previous studies [35]. It is worth noting that the Li^+^ intercalation composite cathode did not destroy the α-MoO_3_ crystal structure, and the surface remained smooth during the discharge process. This further illustrates that the lithium storage mechanism is the topological redox reaction, which is different from the mechanism of the α-MoO_3_/CNTs composite as the anode for the lithium-ion battery. When the α-MoO_3_/CNTs composite was used as an anode electrode for lithium storage in a voltage window of 0.01–3 V, a clear Li_2_O crystalline layer was observed on the surface of the α-MoO_3_ nanobelts, while no obvious Li_2_O crystalline layer was observed on the Li^+^ ion-intercalated α-MoO_3_/CNTs composite cathode after discharge [10]. Amplifying the edge position of the nanobelts in the Li^+^-intercalated α-MoO_3_/CNTs composite cathode, as shown in the HRTEM image of Figure 4b, both 3.64 Å lattice spacing corresponding to the (002) crystal plane of the bent CNTs and 2.07 Å lattice spacing corresponding to the (104) crystal plane of Li_x_MoO_3_ (Li_1.66_Mo_0.66_O_2_) can be seen as a consequence of the insertion of Li^+^ ions into the α-MoO_3_ in the α-MoO_3_/CNTs composite. The element mapping image of Figure 4c indicates that the Mo and C elements from the α-MoO_3_/CNTs composite cathode, from the distribution of P element and EDS dark field image, can be further seen in the composite cathode Li^+^ ion storage primarily through the α-MoO_3_/CNTs composite cathode of α-MoO_3_. According to previous studies, when the α-MoO_3_/CNTs composite is used as an anode electrode, composite-intercalated Li^+^ converted into Mo and Li_2_O, and the Li^+^ ions’ deintercalation removed Li_2_MoO_3_ amorphous based on the conversion reaction [10,35,36]. In the subsequent cycle process, the reversible reaction between Mo and Li_2_O with Li_2_MoO_3_ occurs, which cannot return to the original α-MoO_3_/CNTs composite state, and more capacity loss occurs. Therefore, compared with the lithium storage mechanism of the topological redox reaction as the α-MoO_3_/CNTs composite cathode electrode of the lithium-ion battery, the first coulomb efficiency is lower when used as the anode electrode [9]. Figure 4d–f show the XPS images of the α-MoO_3_/CNTs composite cathode after the first discharge. Figure 4d XPS survey spectrum signifies the Mo, O, and C elements in the α-MoO_3_/CNTs composite, as well as Li, P, and F elements, from the intercalation of Li^+^ into the composite cathode during discharge and the generation of the interface phase between the composite cathode and the electrolyte. Compared with the initial α-MoO_3_/CNTs composite cathode, in the Mo_3_d high-resolution XPS spectrum (Figure 4f), after Li^+^ ions intercalation, the Mo 3d_5/2_ and Mo 3d_3/2_ of the discharged composites shifted to a small angle, which is attributed to the reaction: MoO_3_ + *x*Li^+^ + *x*e^−^ ⟷ Li*_x_*MoO_3_,(1)
and to the formation of Li_x_MoO_3_ [37,38,39]. Not only that, but the Li 1 s high-resolution XPS spectrum (Figure 4g), located in 54.9 eV and 56.7 eV, can be attributed to the formation of α-MoO_3_/CNTs composite cathode and electrolyte interface phases (CEI) and Li^+^ ions intercalation into the α-MoO_3_ layer of the formation. 

The current study conducted cyclic voltammetry tests at different scan rates to explain the lithium storage mechanism of the CNTs added α-MoO_3_/CNTs composite cathode. As shown in Figure 4g–i, the CV curve presented that the redox peak position did not shift significantly, and the curve shape was almost unchanged as the scan rate increased, indicating a better rate performance. When the scan rate was greater than 0.6 mV/s, the charge storage was dominated by the capacitance and less limited by the diffusion-controlled redox reaction, indicating a greater Li^+^ transfer kinetic. And when the scan rate was 1.0 mV/s, the diffusion control was less than 32.7%. The α-MoO_3_/CNTs composite cathode will lead to faster reaction kinetics and an improved rate performance.

The rate capability and cycle performance of the α-MoO_3_/CNTs composite cathode are shown in Figure 5a,b. When the current density was 100 mA/g, the composite cathode had an initial specific capacity of 266 mAh/g. The α-MoO_3_/CNTs composite cathode with 168, 139, 122, 112, and 77 mAh/g of specific capacity at the current density increased to 200, 300, 400, 500, and 1000 mA/g, respectively, and when the current density returned to 100 mA/g, the composite cathode specific capacity returned to about 180 mAh/g. This demonstrates that the composite structure was hardly destroyed as the current density increased. Compared with the pure α-MoO_3_ cathode, when the current density was up to 1000 mA/g, the pure α-MoO_3_ cathode had almost no capacity. Furthermore, when the current density returned to the initial 100 mA/g, the pure α-MoO_3_ cathode specific capacity was much smaller than the initial specific capacity and was only 100 mAh/g. As a cathode electrode, CNTs hardly store Li^+^, but they effectively improve the Li^+^ reversible storage characteristics of α-MoO_3_ in the α-MoO_3_/CNTs composite and greatly improve the rate performance. Figure 5b shows the cycle performance of the α-MoO_3_/CNTs composite cathode and the pure α-MoO_3_ cathode at a current density of 100 mA/g. After 150 cycles, the α-MoO_3_/CNTs composite cathode still had a specific capacity of 93 mAh/g, while the pure α-MoO_3_ cathode had a specific capacity of only 34 mAh/g. The rate and cycle performance tests show that the capacity of the α-MoO_3_ cathode will decay rapidly during the initial cycle process, while it will be relatively stable during the subsequent cycles. This is because the intralayer sites of Li^+^ intercalation in α-MoO_3_ cannot be completely reversible during the initial charge-discharge process. With the gradual occupation of the intralayer sites, the irreversible storage of Li^+^ is improved [40]. The improvement of lithium storage electrochemical performance of α-MoO_3_ by CNTs in the composite cathode was proved; the CNTs embedded into nanobelts to enlarge the interlayer spacing of α-MoO_3_, and the three-dimensional conductive network of CNTs improved the poor conductivity and the structural failure caused by volume expansion of pure α-MoO_3_ during Li^+^ deintercalation. 

To further explore the reason why CNTs improve the electrochemical performance of the α-MoO_3_/CNTs composite cathode, the Li^+^ diffusion kinetics of the composite cathode and pure α-MoO_3_ cathode during charging and discharging were investigated by the galvanostatic intermittent titration technique. As shown in Figure 5c,d, the diffusion coefficient of α-MoO_3_/CNTs composite cathode was always higher than that of the pure α-MoO_3_ cathode, which proves that Li^+^ diffused faster in the α-MoO_3_/CNTs composite cathode. This may have been due to the presence of the CNTs’ three-dimensional conductive network shortening the ion transport path. Meanwhile, we tested the electrochemical impedance spectroscopy for both the α-MoO_3_/CNTs composite cathode and pure α-MoO_3_ cathode in Figure 5e. It shows that compared to pure α-MoO_3_, the semicircle diameter of the α-MoO_3_/CNTs cathode in the high-frequency region was smaller, corresponding to the charge transfer resistance at the interface between the electrode and the electrolyte. The fitting circuit is shown in the upper left of Figure 5e (Rct = 53.33 Ω), and the enhanced conductivity indicates that the α-MoO_3_/CNTs composite cathode had a higher electron conduction efficiency and faster kinetic reaction. In Figure 5f, compared with pure α-MoO_3_, the α-MoO_3_/CNTs composite cathode had a smaller diffusion resistance (Warburg factor, *Z*_w_) in the low-frequency range, that is, the slope of the fitting curves of Z′ and ω^−1/2^ was smaller, which further verifies that the Li^+^ diffusion in the composite cathode was faster. The Li^+^ ion diffusion coefficient can be calculated according to the following formula:(2)$$D_{\mathrm{Li}}=\frac{R^{2}T^{2}}{{2A}^{2}n^{4}F^{4}C^{2}\sigma^{2}}$$

In Equation (2), D_Li_ is the diffusion coefficient of Li^+^ (cm^2^/s), R is the gas constant (8.314 J/Kmol), T is the temperature during the experiment (298 K), A is the electrode material area (cm^−2^), n is the number of electrons per mole of the active material transferred in the electrode reaction, F is the Faraday constant (96,500 C/mol), C is the lithium ion concentration (mol/L), and σ is the slope of Z′~ω^−1/2^. The D_Li_ = 9.3 × 10^−19^ cm^2^/s in the α-MoO_3_/CNTs composite cathode. The CNTs were embedded inside α-MoO_3_ nanobelts and cross-linked between α-MoO_3_ nanobelts, which improved the conductivity of the composite cathode and accelerated the transmission of Li^+^; it also effectively inhibited the volume expansion of α-MoO_3_ during charging and discharging. These tests show that the α-MoO_3_/CNTs composite cathode had faster electron and ion transfer rates and faster reaction kinetics.

## 3. Conclusions

In summary, the CNTs were added by the one-step hydrothermal reaction to synthesize the α-MoO_3_/CNTs composite and thus effectively solve the poor conductivity and cycle stability of pure α-MoO_3_. The part of CNTs was embedded inside the α-MoO_3_ nanobelts, which made the α-MoO_3_ have a larger interlayer spacing and provided more active sites and faster kinetic reaction for lithium storage. Another part of CNTs formed a three-dimensional conductive network between α-MoO_3_ nanobelts, enhancing the electrical conductivity of the composite, accelerating the electron conduction in the energy storage process, shortening the ion transport path, and alleviating the structural fragmentation caused by the volume expansion during the α-MoO_3_ deintercalation of Li^+^, leading to a significantly improved rate performance and cycle life of lithium storage.

## Figures and Tables

**Figure 1 nanomaterials-13-02272-f001:**
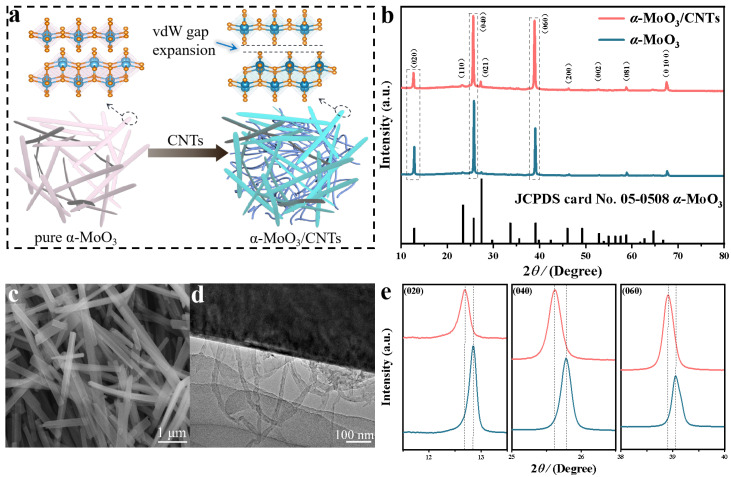
(**a**) Schematic of the comparison of α-MoO_3_/CNTs composite and pure α-MoO_3_. (**b**) XRD patterns of the α-MoO_3_/CNTs composite and pure α-MoO_3_. (**c**) SEM image of the α-MoO_3_/CNTs composite. (**d**) TEM image of the α-MoO_3_/CNTs composite. (**e**) XRD patterns around the (020), (040), and (060) diffraction peaks.

**Figure 2 nanomaterials-13-02272-f002:**
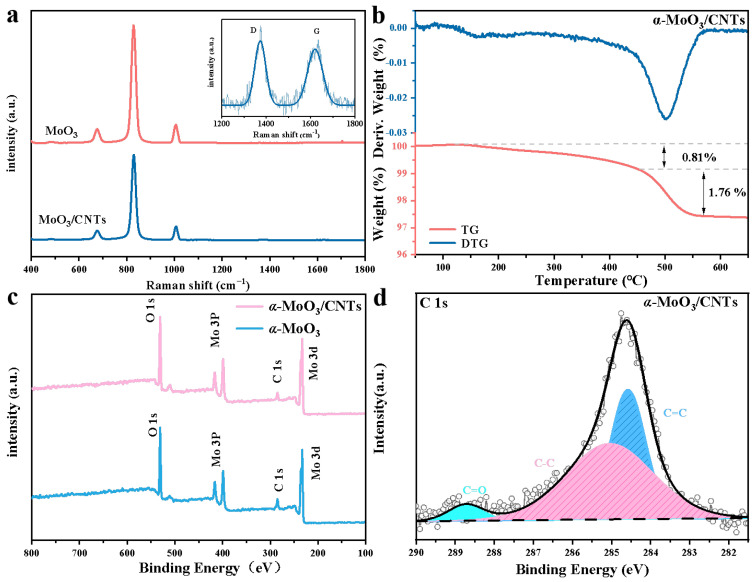
(**a**) Raman spectra of the α-MoO_3_/CNTs composite and pure α-MoO_3_. (**b**) TG and DTG curve of the α-MoO_3_/CNTs composite. (**c**) Survey XPS spectra of the α-MoO_3_/CNTs composite and pure α-MoO_3_. (**d**) The high-resolution XPS spectra of the C 1s region of the α-MoO_3_/CNTs composite.

**Figure 3 nanomaterials-13-02272-f003:**
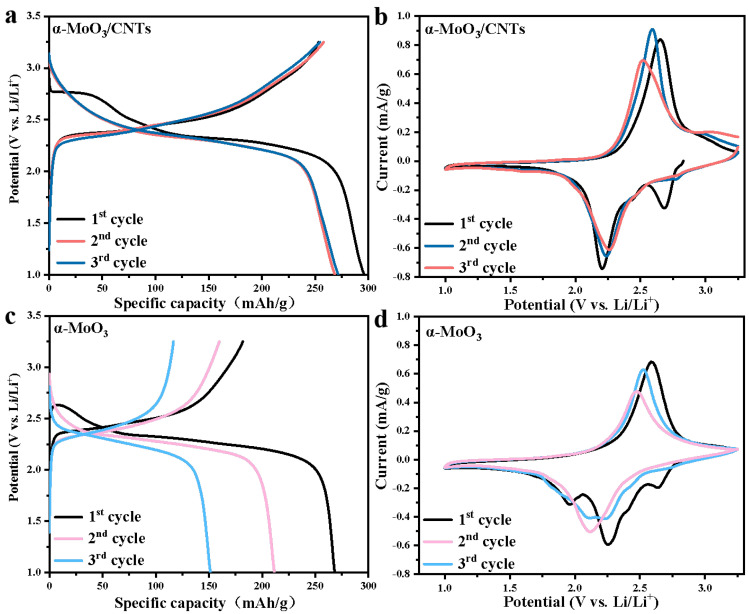
(**a**) GCD curves of the α-MoO_3_/CNTs composite cathode. (**b**) CV curves of the α-MoO_3_/CNTs composite cathode. (**c**) GCD curves of the pure α-MoO_3_ cathode. (**d**) CV curves of the pure α-MoO_3_ cathode.

**Figure 4 nanomaterials-13-02272-f004:**
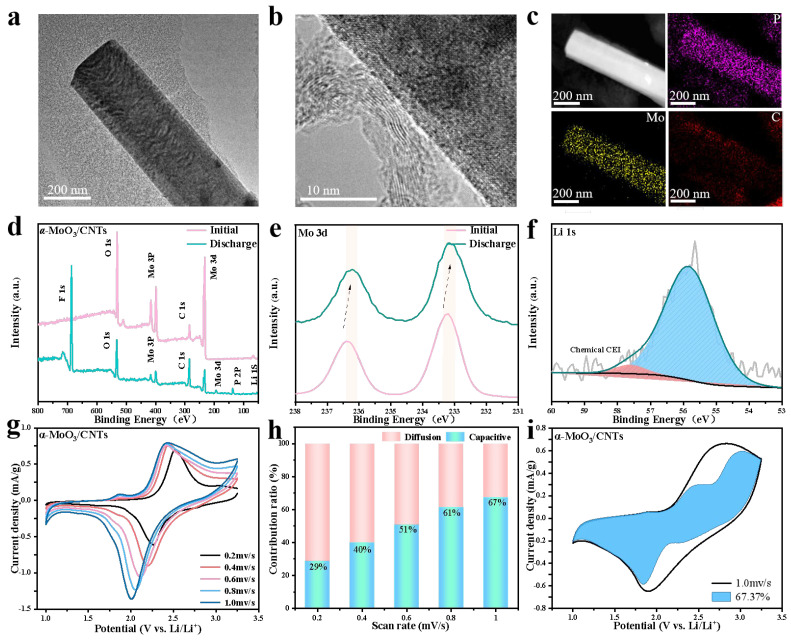
The Li^+^ ion-intercalated α-MoO_3_/CNTs composite cathode (**a**) TEM image, (**b**) HRTEM image, (**c**) EDS element images, (**d**) Survey XPS spectrum, (**e**) Mo 3d, and (**f**) Li 1s spectra. (**g**) CV curves of the α-MoO_3_/CNTs composite cathode were obtained. (**h**) The normalized charge contribution of the capacitive and diffusion-controlled capacities was extracted. (**i**) Capacitive charge storage contribution (blue area) to the total capacity at a scan rate of 1 mV/s.

**Figure 5 nanomaterials-13-02272-f005:**
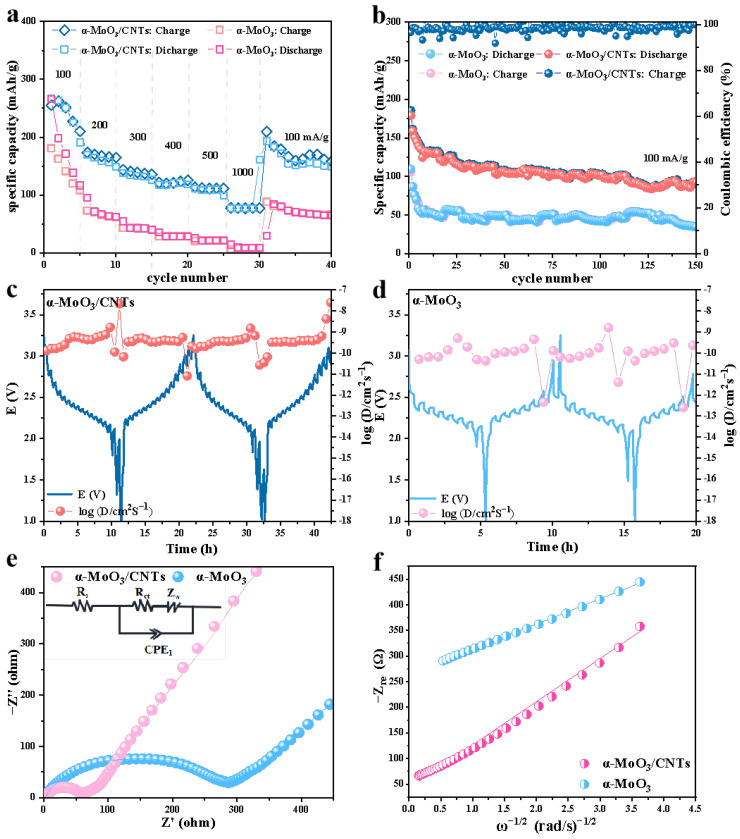
(**a**) The rate capability of the α-MoO_3_/CNTs composite and pure α-MoO_3_ at different current densities. (**b**) Cycling performance of the α-MoO_3_/CNTs composite and pure α-MoO_3_ at 100 mA/g. GITT curve and diffusion coefficients calculated from GITT curves for (**c**) the α-MoO_3_/CNTs composite cathode and (**d**) pure α-MoO_3_ cathode. (**e**) Nyquist plots of the α-MoO_3_/CNTs composite and pure α-MoO_3_. (**f**) The relationship plots between Z’ and ω^−1/2^ in the low-frequency range of the α-MoO_3_/CNTs composite cathode.

**Table 1 nanomaterials-13-02272-t001:** The literature data of the α-MoO_3_/carbon composite cathode.

Cathode	Initial Capacity	Rate: Capability/Current Density	Cycling Life	Ref.
carbon-coated MoO_3_	258 mA h g^−1^	118 mAh g^−1^/3 A g^−1^	125 mAh g^−1^ at 1.5 A g^−1^ after 500 cycles	[13]
α-MoO_3−x_ plasma etching	224.2 mA h g^−1^	≈90 mAh g^−1^/5 A g^−1^	67.3 mAh g^−1^ at 1 A g^−1^ after 1000 cycles	[33]
α-MoO_3_/SWCNT-COOH	193.8 mA h g^−1^	--	70 mAh g^−1^ at 0.5 A g^−1^ after 117 cycles	[22]
α-MoO_3_/N-CNTs	250 mA h g^−1^	190 mAh g^−1^/0.3 A g^−1^	250 mAh g^−1^ at 0.3 A g^−1^ after 50 cycles	[11]
α-MoO_3_/PEO	352 mA h g^−1^	--	124 mAh g^−1^ at 0.03 A g^−1^ after 50 cycles	[34]
This work	296 mA h g^−1^	77.2 mAh g^−1^/1 A g^−1^	93 mAh g^−1^ at 0.1 A g^−1^ after 150 cycles	This work

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
