# Peer review of "Enhanced Lithium Storage Performance of α-MoO3/CNTs Composite Cathode"

_nanomaterials, 2023, doi:10.3390/nano13152272_

Round 1

Reviewer 1 Report

The manuscript of Ang Gao, Dawei Sheng, Xiaoxu Li and Qiang Zhang describes the complex research on the physico- and electrochemical characteristics of α-MoO3/CNTs composite prepared by one-step hydrothermal method, which can be used as cathode material in Li-ion batteries.

The paper is well written, well-structured and its subject matter is within the general aim and scope of the Nanomaterials journal. The introduction provides sufficient background on the subject and contains general information about Li-ion batteries, cathode materials, properties and future prospects of MoO3. All relevant references are also included.

The adequate and broad characterization of considered materials is provided. The whole study is well designed. In addition, the results are clearly presented, the discussion is given in a logical way and the conclusions are supported by the results.

Taking all things into consideration, the performed and displayed study is characterized by high level of novelty as well as high importance in the field of energy storage systems. Nevertheless, please clarify whether the diffusion coefficient was determined based on the Weber factor and not the Warburg? Therefore, I recommend it for publication after deliberating the point mentioned above and those listed below.

Minor issues:

· The superscript should be used in the “Li+” expressions throughout the text.

· Page 2, line 92 in the PDF file: “the XRD spectra” should be replaced by “the XRD pattern”.

· Page 2, line 93 in the PDF file: probably a dot is missing between “Figure 1e” and “Compared”. An analogous remark applies to the 112 line.

· Please consider changing the term "discharge platform" in the manuscript.

· The inset in Figure 2a is not legible.

Minor editing of English language required.

Author Response

Response

According to the suggestions, we have made revisions to the manuscript as follows:

We referred to the impedance spectra (EIS) results in the existing literature1, 2 and checked and modified the description of the AC impedance testing in the text. In Figure 5f, we found that the α-MoO3-x/CNTs composite cathode has a smaller diffusion resistance (Warburg factor, Zw) in the low-frequency range when compared with pure α-MoO3. We Refitted the slope of Z' and ω-1/2 to calculate the diffusion coefficient of lithium ion. Thank you very much for your valuable advice.

  1. We have made the necessary corrections to all instances of "Li+" in the text and the corresponding parts in the manuscript have been revised one by one and marked red in the manuscript.
  2. Modify "the XRD spectra" to "the XRD pattern" on page 2, line 92 of the manuscript.
  3. In the manuscript, page 2, line 93: A supplement is provided with the missing punctuation points between "Figure 1e" and "Compared", and in line 112. The use of full-text punctuation was also checked.
  4. We have made the necessary corrections to the manuscript, changing "discharge platform" to "discharge voltage plateaus" and "voltage plateaus", etc. These corrections have been annotated in red.
  5. Change the color matching and font size of the illustration in Figure 2a to make it clearer on page 2. 

References

  1. Zhou, Y.; Wang, X.; Shen, X.; Shi, Y.; Zhu, C.; Zeng, S.; Xu, H.; Cao, P.; Wang, Y.; Di, J.; Li, Q. Journal of Materials Chemistry A 2020, 8, (23), 11719-11727.
  2. Zeng, T.; Liu, X.; Kang, W.; He, H.; Zhang, J.; Li, X.; Zhang, C. Angew Chem Int Ed Engl 2021, 60, (50), 26218-26225.

Reviewer 2 Report

The manuscript can be published after slight improvement. Please, find below my suggestion and comments

1/ Please, put the exeperimental part to the manuscript. P

2/ Please, describe the synthesis procedure.

3/ Please, give the surface area of obtained materials.

4/ Please, compare the obtained capacity with last published results. The provided parameters are not excellent ( α-MoO3/CNTs composite cathode still retained 93 mAh/g capacity.) and the title of presented manuscript doesn not show the described situation. The obtained parameters should be compared with key works (and now only authors previous publication).

5/ What kind of CNT the authors used for synthesis?

6/ Please, clarify how big is theoretical limit of Li+ /Mo for such composition?

Minor editing of English language required. Please, check the punktuation and spelling

Author Response

We accept these comments. According to the suggestions, we make revisions in manuscript as follows. Adding figures are given in the word files.

  1. We have moved the experimental part to the supporting information section.
  2. The nitrogen adsorption/desorption curves and pore size distribution curves of α-MoO3/CNTs are shown in Fig. a and b, respectively. The H3-type hysteresis loop in Fig. a shows that α-MoO3/CNTs belongs to the type-IV isotherm, which is a typical porous material. Moreover, the specific surface area of α-MoO3/CNTs is 933 m2/g.
  3. We have conducted a comparison of our results with those published in similar materials. The comparison is presented in the table below

cathode

Initial capacity

Rate:capability/current density

Cycling life

Ref.

carbon-coated MoO3

258 mA h g-1

118 mAh g-1/3 A g-1

125 mAh g-1 at 1.5 A g-1 after 500 cycles

1

α-MoO3-x plasma etching

224.2 mA h g-1

≈90 mAh g-1/5 A g-1

67.3 mAh g-1 at 1 A g-1 after 1000 cycles

2

α-MoO3/SWCNT-COOH

193.8 mA h g-1

--

70 mAh g-1 at 0.5 A g-1 after 117 cycles

3

α-MoO3/N-CNTs

250 mA h g-1

190 mAh g-1/0.3 A g-1

250 mAh g-1 at 0.3 A g-1 after 50 cycles

4

α-MoO3/PEO

352 mA h g-1

--

124 mAh g-1 at 0.03 A g-1 after 50 cycles

5

This work

296 mA h g-1

77.2 mAh g-1/1 A g-1

93 mAh g-1 at 0.1 A g-1 after 150 cycles

This work

(In this table, the rate refers to the specific capacity obtained from the maximum current density test described in the text.)

  1. In the manuscript, we used hydrophobic multiwall carbon nanotubes (CNTs). To facilitate the synthesis of α-MoO3/CNTs composite with an internal embedded structure, we treated the CNTs to make them rich in functional groups. Specifically, we acidified the multiwall CNTs to make them hydrophilic

6.A small amount of CNT in this composite cannot be used as a positive electrode for lithium ions. According to previous research and a review article6, MoO3 has been shown to have a maximum theoretical energy density of 930 Wh kg-1 (372 mAh g-1 × 2.5 V vs. Li/Li+), which is significantly higher than that of current commercial cathode materials such as LiCoO2 (~580 Wh kg-1) and LiFePO4 (~500 Wh kg-1).

Intercalation reactions occurring at the positive pole of the α-MoO3/CNTs complex:

MoO3 + xLi+ + xe- ↔ LixMoO3, 0 < x ≤ 2, with a two-electron reaction (Mo6+/Mo4+)7,8.

References

  1. Hu, Z.; Zhang, X.; Peng, C.; Lei, G.; Li, Z. Journal of Alloys and Compounds 2020, 826.
  2. Zhang, G.; Xiong, T.; Yan, M.; He, L.; Liao, X.; He, C.; Yin, C.; Zhang, H.; Mai, L. Nano Energy 2018, 49, 555-563.
  3. Mendoza-Sánchez, B.; Grant, P. S. Electrochimica Acta 2013, 98, 294-302.
  4. Zhang, H.; Liu, X.; Wang, R.; Mi, R.; Li, S.; Cui, Y.; Deng, Y.; Mei, J.; Liu, H. Journal of Power Sources 2015, 274, 1063-1069.
  5. Nadimicherla, R.; Chen, W.; Guo, X. Materials Research Bulletin 2015, 66, 140-146.
  6. Sheng, D.; Liu, X.; Zhang, Q.; Yi, H.; Wang, X.; Fu, S.; Zhou, S.; Shen, J.; Gao, A. Batteries & Supercaps 2023, 6, (5), e202200569.
  7. Hashem, A. M.; Askar, M. H.; Winter, M.; Albering, J. H.; Besenhard, J. O. Ionics 2007, 13, (1), 3-8. Baldoni, M.; Craco, L.; Seifert, G.; Leoni, S. Journal of Materials Chemistry A 2013, 1, (5), 1778-1784.

Reviewer 3 Report

Paper has no experimental section, authors must at least read their manuscript before submitting. Nanomaterials is a reputed scientific journal. Not suitable for publication. Reject.

Author Response

We accept these comments. According to the suggestions, we make revisions in manuscript as follows,

  1. We placed the experimental part in the supporting information.

Round 2

Reviewer 2 Report

The manuscript can be published at Nanomaterials.

I suggest to include the surface area of material (it is surprisingly low) and the table of comparison to main text (answers to my question 2 and 3). It should be interesting to readers of this paper! 

Author Response

Thanks for your comments, the specific surface area of our cathode material is indeed a bit low, but in past reports, as shown below, MoO3 and MoO3/CNTs as electrode materials, with a specific surface area of 0.849 m2/g and 19.8 m2/g, respectively[1]. The specific surface area of the MoO3 electrode prepared in 5 M HNO3 is 16.7 m2/g[2]. MoO3NPs as the electrode material, its specific surface area is 3.1299 m2/g[3]. This paper has similar results to these tests. In contrast, the present paper utilizes MoO3 components in composites, topology-based redox reactions, the intercalation reaction mechanism that occurs, as the positive electrode for lithium storage, this mechanism has faster charge storage dynamics and shows the pseudocapacitance characteristics of charging in several seconds[4]. But the low specific surface area defects of our composites still need to be overcome to bring about improved electrochemical properties. Thank you sincerely again for your comments. [1] Kiran L., Aydinol M. K., Ahmad A., Shah S. S., Bahtiyar D., Shahzad M. I., Eldin S. M. and Bahajjaj A. A. A., Flowers Like alpha-MoO(3)/CNTs/PANI Nanocomposites as Anode Materials for High-Performance Lithium Storage, Molecules, 2023, 28(8).https://doi.org/10.3390/molecules28083319 [2] Han B., Lee K.-H., Lee Y.-W., Kim S.-J., Park H.-C., Hwang B.-M., Kwak D.-H. and Park K.-W., MoO3 Nanostructured Electrodes Prepared via Hydrothermal Process for Lithium Ion Batteries, International Journal of Electrochemical Science, 2015, 10(5): 4232.https://doi.org/10.1016/s1452-3981(23)06618-x [3] Sumedha H. N., Shashank M., Praveen B. M. and Nagaraju G., Electrochemical activity of ultrathin MoO3 nanoflakes for long cycle lithium ion batteries, Results in Chemistry, 2022, 4: 100493.https://doi.org/10.1016/j.rechem.2022.100493 [4] Li Y., Sun H., Cheng X., Zhang Y. and Zhao K., In-situ TEM experiments and first-principles studies on the electrochemical and mechanical behaviors of α-MoO3 in Li-ion batteries, Nano Energy, 2016, 27: 95.https://doi.org/10.1016/j.nanoen.2016.06.045
